# Automated pipeline framework for processing of large-scale building energy time series data

Arash Khalilnejad[1,5], Ahmad M. Karimi[2,5], Shreyas Kamath[1,5], Rojiar Haddadian[2,5], Roger H. French[2,4,5]*, Alexis R. Abramson[3,6¤]

1 Department of Electrical, Computer, and Systems Engineering, Case School of Engineering, Case Western Reserve University, Cleveland, Ohio, United States of America, 2 Department of Computer and Data Sciences, Case School of Engineering, Case Western Reserve University, Cleveland, Ohio, United States of America, 3 Department of Mechanical and Aerospace Engineering, Case School of Engineering, Case Western Reserve University, Cleveland, Ohio, United States of America, 4 Department of Materials Science and Engineering, Case School of Engineering, Case Western Reserve University, Cleveland, Ohio, United States of America, 5 SDLE Research Center, Case School of Engineering, Case Western Reserve University, Cleveland, Ohio, United States of America, 6 Great Lakes Energy Institute, Case School of Engineering, Case Western Reserve University, Cleveland, Ohio, United States of America

¤ Current address: Thayer School of Engineering at Dartmouth, Hanover, New Hampshire, United States of America
* roger.french@case.edu

**Data Availability Statement:** The data underlying the results presented in the study are available from https://osf.io/unm43/, DOI: 10.17605/OSF.IO/UNM43.

## Abstract

Commercial buildings account for one third of the total electricity consumption in the United States and a significant amount of this energy is wasted. Therefore, there is a need for "virtual" energy audits, to identify energy inefficiencies and their associated savings opportunities using methods that can be non-intrusive and automated for application to large populations of buildings. Here we demonstrate virtual energy audits applied to large populations of buildings' time-series smart-meter data using a systematic approach and a fully automated Building Energy Analytics (BEA) Pipeline that unifies, cleans, stores and analyzes building energy datasets in a non-relational data warehouse for efficient insights and results. This BEA pipeline is based on a custom compute job scheduler for a high performance computing cluster to enable parallel processing of Slurm jobs. Within the analytics pipeline, we introduced a data qualification tool that enhances data quality by fixing common errors, while also detecting abnormalities in a building's daily operation using hierarchical clustering. We analyze the HVAC scheduling of a population of 816 buildings, using this analytics pipeline, as part of a cross-sectional study. With our approach, this sample of 816 buildings is improved in data quality and is efficiently analyzed in 34 minutes, which is 85 times faster than the time taken by a sequential processing. The analytical results for the HVAC operational hours of these buildings show that among 10 building use types, food sales buildings with 17.75 hours of daily HVAC cooling operation are decent targets for HVAC savings. Overall, this analytics pipeline enables the identification of statistically significant results from population based studies of large numbers of building energy time-series datasets with robust results. These types of BEA studies can explore numerous factors impacting building energy efficiency and virtual building energy audits. This approach enables a new generation of data-driven buildings energy analysis at scale.

**Funding:** This work was supported by the U.S. Department of Energy, Advanced Research Projects Agency-Energy (ARPA-E), under award DE-AR-0000668. All the authors were funded or partially funded by this award. The detailed information is given in: https://arpa-e.energy.gov/technologies/projects/virtual-building-energy-audits.

**Competing interests:** Case Western Reserve University (CWRU) is the owner of inventions associated with this work being commercialized by Edifice Analytics Inc., and as such, could gain royalties. ARA and RHF invented intellectual property, have a patent application and are board members of Edifice Analytics. ARA, RHF and AK have ownership interest in Edifice Analytics. This does not alter our adherence to PLOS ONE policies on sharing data and materials.

## Introduction

Buildings account for approximately one-third of the world's total electricity consumption [1]. In the United States, commercial buildings account for 36% of the total energy consumption, of which approximately 30% is wasted [2]. Hence, reducing wastage of energy and improving the efficiency of buildings' energy consumption has significant importance [3]. Due to the cost and time required for conventional on-site building energy audits, "virtual" energy audits, which has no need for setting foot in a building and use appropriate diagnostic and prognostic tools, is an important goal of building research. We have developed a virtual energy audits tool, EDIFES (Energy Diagnostics Investigator for Efficiency Savings) using a data-driven analytical approach based on smart-meter data provided by electrical utility companies or building owners [4–8].

Another important challenge for the large scale application of virtual energy audits is the ability to analyze large numbers of buildings and volumes of building time-series data so that energy savings across distinct building populations can be prioritized and buildings can be compared and ranked [9, 10]. However, scalable time series data processing and classification is constrained by the computational demands of some state of the art analytical methods [11–13]. Therefore, not only is an advanced high performance computing cluster essential, but a robust job scheduling pipeline that can automatically ingest, process and analyze the datasets and rank-order and compare the results is also required. For this purpose, NoSQL databases address the dataset scalability issues of peta-byte scale analyses and have seen increasing use in energy research [14]. Distributed computing, including distributed job processing and NoSQL databases, with their intrinsic scalability to large dataset sizes can cope with the computational demands of large datasets and populations efficiently, thereby addressing the inadequacies of traditional relational database management systems (RDBMS) such as SQL databases [15]. NoSQL database management systems such as Cassandra, MongoDB, Redis and HBase can handle query and processing issues of large-scale energy time series datasets [16]. Additionally, for analyzing large datasets with spatial and temporal dimensions, cluster distributed computing tools such as the Hadoop framework and its Hadoop Distributed File System (HDFS) are commonly used to distribute and parallelize computations on a cluster [17]. Studies show that the HBase model, an open-source, non-relational data warehouse that runs on top of HDFS, if implemented properly, could be very efficient for machine learning applications in large-scale energy time series datasets [18].

Scalable data warehousing has shifted the direction of building science and building data analytics, from the evaluation of individual buildings in observational studies, to cross sectional studies using statistically significant populations of buildings. Examples of these new approaches include a recent study on office buildings, where a ranking system was proposed based on occupant behavior using two level K-means clustering [19]. In another case, energy use intensity (EUI) was used as the basis for clusters with an outlier detection method prior to analysis [20]. In a third recent study by Wilcox et al., a big data platform for ingesting and analyzing smart-meter data was presented [21]. They introduced various requirements and infrastructure necessary for efficient big data evaluation of smart-meter datasets named as smart-meter analytics scaled by hadoop (SMASH) which can process datasets at a 20 TB scale. These distributed computing "big data" analytical approaches to building research have been applied to processing and analyzing building energy datasets, to building population studies, and to the use of machine learning methods [22]. Generally, the development of data-driven analysis methods has lead to the introduction of new savings opportunities on the energy consumption side, whereas the innovations on savings in energy generation such as renewable energy implementation and storage has already been discussed in several research studies [23–27].

A robust framework for large scale dataset processing requires not only an advanced compute infrastructure but also a robust analytics pipeline, to enable efficient and precise data processing [28]. For example in building time series data, issues such as meter malfunctions and data manipulation errors, lead to anomalous data values; and this requires accurate time series anomaly detection and remediation methods [29]. Even the data structure, data query, and output results are important for a generalized and efficient framework that can analyze diverse building datasets and assemble them with essential other datatypes such as weather data and system meta-data [30]. Studies of statistically significant samples or populations of buildings can transform building science from an observational science to one based on a statistically sound foundation [31–35]. In a data-driven building energy study of urban buildings in Stockholm, it was estimated that 5532 buildings could have savings through retrofitting, with potential improvement in peak electric power of 147 MW [36]. Obviously, studies like this are enabled through compute automation, data analysis pipelines, and high performance and parallel computing infrastructure.

Heating, Ventilation and Air Conditioning (HVAC) systems are one of the largest contributors to building energy consumption in commercial buildings and data-driven energy analysis can assess their energy consumption and efficiency [37–41]. The development of the Building Energy Analytics (BEA) Pipeline can help identify buildings and building use types with the highest energy savings opportunities. Other than savings opportunities in a building, identifying the schedule, rescheduling, setpoint setback, and controlling the auxiliary units and HVAC operation can also help electricity grid operations through, for example, peak load reduction [42]. However, due to lack of equipment level data, a data-driven study of the HVAC operational scheduling and energy efficiency of a large-scale population of buildings, has not been possible up to this point.

In this paper, we demonstrate the development of an energy analytics pipeline wherein data, after being queried, automatically passes through multiple preprocessing, cleaning, assembly, and ingestion steps in a high performance and parallel computing environment with fast-track, smart, and interactive capabilities. The objective of this analytics automation is to process files by performing all the necessary actions with minimal manual intervention, well informed by the potential issues associated with energy datasets and building operations. With a case study over 816 commercial buildings, we will demonstrate how the introduced pipeline enables in-depth understanding of HVAC cooling performance and operational time.

## Methods

### Building dataset

**Time series.** Building-energy time-series datasets are electricity consumption data with timestamps that capture many aspects of the behavior and performance of a building, as is shown for one building over a one year period in Fig 1. This dataset has gone through a series of preprocessing and data cleaning steps that are discussed here. For a cross-sectional study of a population or sample of tens or hundreds of buildings, a number of challenges arise. First, the building energy datasets should have a single, common data structure, independent of their original source. Second, they should be "cleaned" to address issues such as anomalies, outliers, missing data points, gaps in the time series, or inconsistencies in the time interval or temporal data point spacing. Third, dataset sizes can become very large, as a function of the time interval, which can vary from 15-minute interval data to sub-minute interval time-series. The design and configuration of the compute infrastructure (high performance and/or distributed computing types) must integrate well with the analytics pipeline developed. Fourth, data warehousing is required not only for storing the data and its metadata, but also to allow the

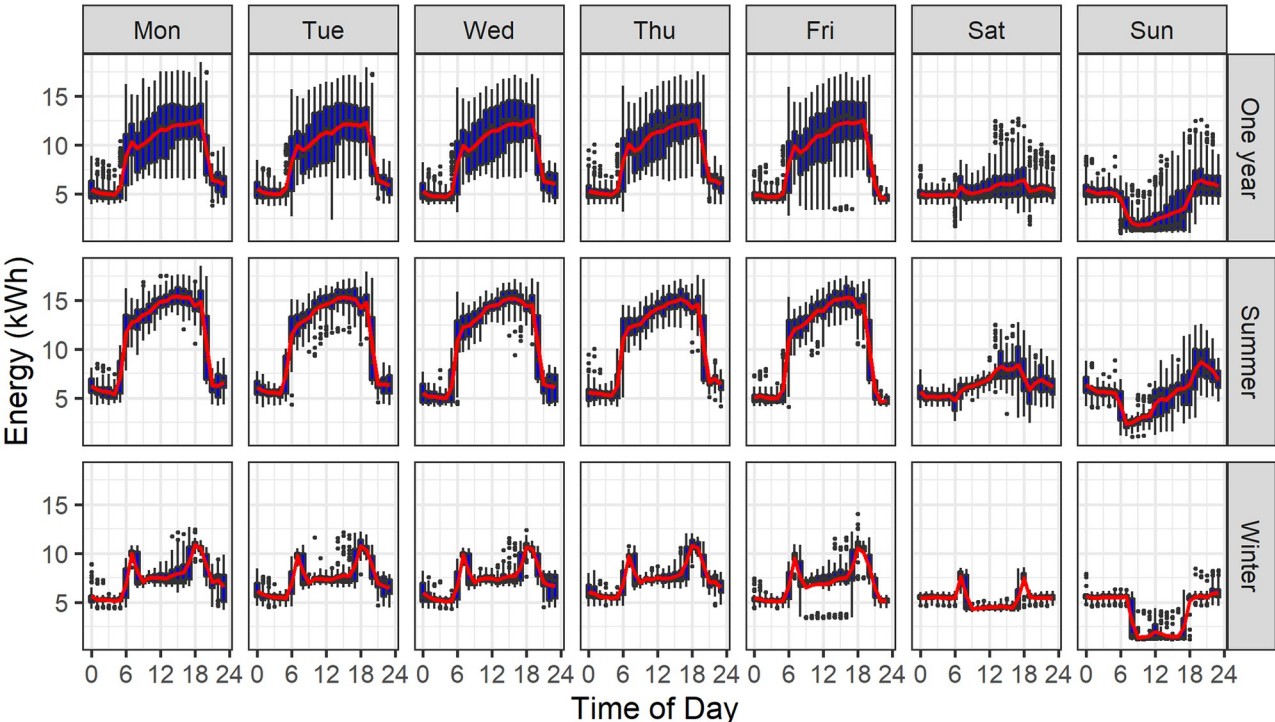

**Fig 1. Energy pattern snapshot.** Time series representation of the characteristic energy consumption of a building from Monday through Friday in the past full year and during the summer (June, July, and August) and winter (December, January, and February) months. The blue vertical boxes show the distribution (middle 50% variability) of energy consumption for the given hour across each season. The whiskers indicate the minimum to maximum consumption, excluding outliers.

storage of full and complete results of the analysis, so as to enable population-based meta-analysis across buildings.

**Population of buildings.** In this study we analyze a population of 816 buildings' energy datasets, whose characteristics are summarized in Fig 2. The buildings are classified into 10 building use types and their datasets are at least one year in duration and have time intervals ranging from one to 60 minutes. The buildings are located across the United States and correspond to a variety of distinct climate zones as specified by the Köppen-Geiger (KG) Climate Zone schema [43–45].

## Pipeline of data acquisition to analysis

The automated process which implies to automatic flow of building data to each step of the pipeline, is designed to do all the required steps for data processing and analytics automatically with an efficient framework and unified structure, and generate comparable results with data from different sources and formats. As shown in Fig 3 the automated process includes the steps for data acquisition, preprocessing, cleaning, ingestion, weather data acquisition and ingestion, meta data processing and ingestion, building energy analytics and reporting. Our developed pipeline enables any applied analysis to be distributed to all the datasets and results stored in the database, automatically.

**Data acquisition.** Building datasets are provided in different formats such as csv, xml, txt, etc. with plenty of variation in their structure. The implementation of the pipeline transforms the heterogeneous datasets to structured ones. As the developed pipeline is automated, the

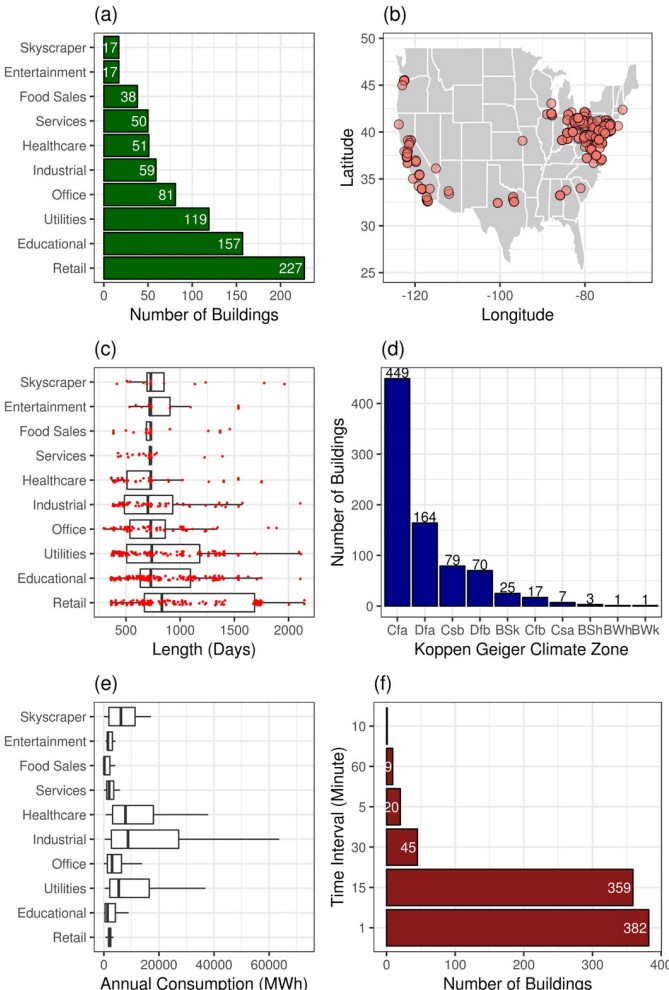

**Fig 2. Population of buildings.** (a) breakdown of the buildings by building use type, (b) location of the buildings, (c) distribution of time length of the dataset for each building use type (d) distribution of buildings by KG climate zone (e) annual consumption distribution by type, and (f) time interval breakdown of dataset.

challenging issue in the pipeline is handling data with unknown structures, column names, units, timestamps, etc. The following steps in the pipeline identify and fix those issues.

**Preprocessing.**

- Tidying: The desired data structure for a building-energy time-series dataset should be a column of POSIXct timestamps with proper structure and time zone, and associated columns of energy consumption and other relevant variables such as temperature, and solar irradiance of the local weather. Yet the smart-meter data is rarely of this format to start with, so a tidying process converts the untidy data to the desired structure.

- Restructuring: In this step, first, the timestamps are fixed, including splining the time series when the timestamps have non-uniform time intervals [46], and translating from UTC time zone to the local timezone. Then, missing points in energy consumption are flagged in a column with logical values, where 1 denotes a missing value while 0 denotes a non-missing value. Finally, columns derived from timestamps and energy consumption, such as day of week, business or non-business days, sunrise and sunset time, etc., are generated. At this

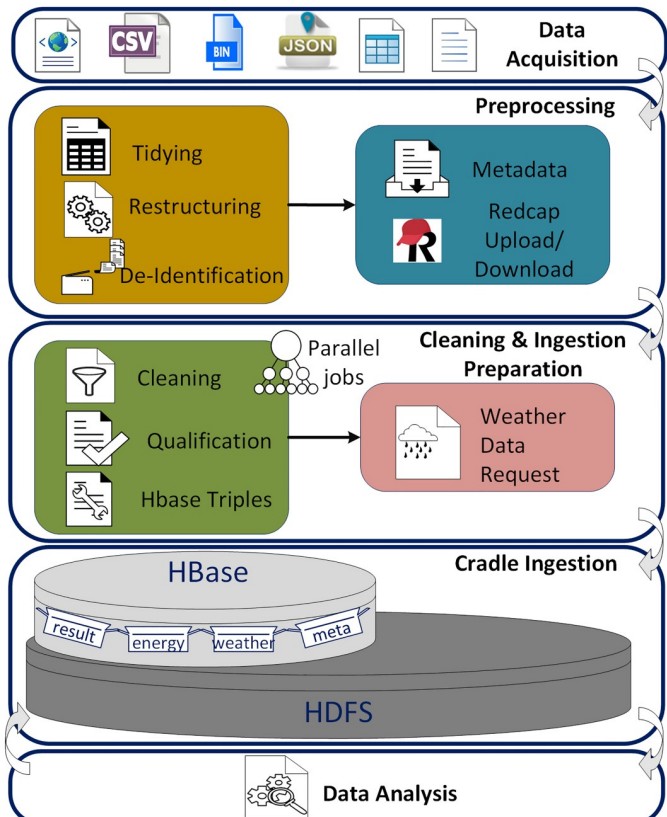

**Fig 3. Energy data processing pipeline.** The pipeline includes data acquisition, typically with differing data structures and file formats, preprocessing for providing a unique data structure and de-identification, cleaning which checks, and removal of anomalous data to improve data quality and prepares HBase triples, cradle ingestion of those triples into the data warehouse, then followed by analysis in HBase.

point, all data from any source are transformed into a consistent structure. Table 1 indicates an example of structured data. For consistency, all column names are 4 digit characters. The description of column names is provided in Table 2.

- De-Identification: Due to inconsistency in file names and privacy and security issues, the data are de-identified with random alphanumeric names.

**Data cleaning.** Since the dataset quality is essential for accurate analytical results, an automated data cleaning process is used for data cleaning including data entry, measurement instrument and data integration errors. The variables in the dataset include quantitative, categorical, postal, and identifier variables from multiple data sources, and can have distinct data

**Table 1. Energy time series data structure.**

| tmst | cons | ecoi | pwdm | bzdy | dywk | wkdn | snrs | nmtm | clen | anfl | forc | amfl |
|---|---|---|---|---|---|---|---|---|---|---|---|---|
| 2016-10-15 00:00:00 | 6.65 | 0 | NA | 1 | Wed | day | NA | 0.00 | 6.65 | 0 | 6.65 | 0 |
| 2016-10-15 00:15:00 | 6.49 | 0 | -0.16 | 1 | Wed | day | NA | 0.25 | 6.49 | 0 | 6.49 | 0 |
| 2016-10-15 00:30:00 | 6.75 | 0 | 0.26 | 1 | Wed | day | NA | 0.50 | 6.75 | 0 | 6.75 | 0 |
| 2016-10-15 00:45:00 | 6.46 | 0 | -0.29 | 1 | Wed | day | NA | 0.75 | 6.46 | 0 | 6.46 | 0 |

**Table 2. Column names and description of building energy dataset.**

| column | column name | format | description |
|---|---|---|---|
| tmst | timestamp | Posixct | local time |
| cons | energy consumption | numeric | in kwh |
| ecoi | energy consumption | 0 or 1 | if 0, actual, |
| | imputation flag | | if 1, missing. |
| pwdm | power demand | numric | diff of energy consumption |
| bzdy | business day | 0 or 1 | if 1, business day, |
| | | | if 0, non-business day. |
| dywk | day of week | character | three letter abbreviation |
| wkdn | week day or end | character | if day, weekday, |
| | | | if end, weekend. |
| snrs | sunrise or sunset | character | if rise, sunrise, |
| | | | if set, sunset, else, NA. |
| nmtm | number of time | numeric | time of day in numeric value |
| clen | cleaned energy | numeric | data with anomalies detected |
| anfl | anomaly flag | 0 or 1 | if 0, actual, |
| | | | if 1, anomaly. |
| forc | energy with forecasted values | numeric | missings and anomalies imputed with forecast |
| amfl | anomaly and missing flag | 0 or 1 | if 0, actual, |
| | | | if 1, missing or anomaly. |

cleaning challenges such as single datapoint or sequential chunks of missing data points, and timestamp merging, data structure, and redundant values issues. To address these issues, we developed a data quality assessment and qualification tool and analysis pipeline.

**Anomaly detection.** To detect different types of the time series outliers [47], first we use classical time series decomposition. Then anomaly detection is applied to the remainder component of the time series and anomalies are removed. At this point the remainder, trend, and seasonal components are recombined to produce a corrected time-series dataset without outliers. In addition, single missing datapoints are imputed by linear interpolation. Due to the possibility of different behavior of the energy consumption pattern during weekends, we apply the anomaly detection algorithm to the dataset for weekdays and weekends, separately.

**Data qualification.** The data quality of the building energy dataset is determined using an A to D grading system based on quality metrics, as summarized in Table 3. The final assessment ("P" for pass, or "F" for fail) requires at least 1 year of good quality time-series data to enable time-series analysis. For example, a building energy dataset with grade of *ACBP* means that it has an anomaly rate of less than 5%, missing data percentage of 15 to 20%, the largest gap of 120 to 164 hours and, it is more than a year long. The ultimate goal of the data qualification tool is to make sure that the data quality is *AAAP* and, if not, to try and transform it to this grade as much as possible.

**Table 3. Data quality grading criteria.**

| | Anomalies (%) | Missing percentage (%) | Largest Gap (Hours) |
|---|---|---|---|
| *A* | Below 5 | Below 10 | Below 120 |
| *B* | 5 to 7 | 10 to 15 | 120 to 164 |
| *C* | 7 to 10 | 15 to 20 | 164 to 240 |
| *D* | Above 10 | Above 20 | Above 240 |

**Abnormal days detection.** Other than those discussed earlier, some anomalies represent abnormal daily energy consumption due to a significantly different consumption pattern compared to other days, defined here as "abnormal days". By a hierarchical clustering algorithm [48], that uses daily time series energy consumption, the abnormal days with extremely high or low consumption or irregular pattern are identified. The irregular pattern in daily energy consumption corresponds to days with significantly different energy consumption curves compared to other days. The clustering algorithm computes the euclidean distance of corresponding energy data points of different days, and clusters based on the similarity of the euclidean distance of days.

**Data assembly.** In addition to the energy consumption data, weather data and the building metadata are queried and assembled with the energy data. This data assembly is critical for automation of the building analysis pipeline.

**Weather data.** Weather data is obtained from the SolarGIS cloud [49]. SolarGIS uses satellite imagery in combination with a quantitative atmospheric model to produce ground-level weather data for the United States on a 3.5 km pixel size and 30-minute time interval. The weather variables include temperature, relative humidity, solar global horizontal irradiance, and the UTC timestamp. Given the building's location (longitude, latitude, or zipcode) and the start and end time of its time series, we submit a SolarGIS API (application programming interface) request, translate the timestamps of the response to the local time zone, and ingest this to the weather table stored in HBase [50]. Storing weather data in a dedicated HBase weather table, allows us to perform a local query to check if we already possess the needed weather data, prior to making a SolarGIS query. At the point of building energy analysis, the weather data timestamps are splined to match the energy data timestamps and merged with energy dataset. Table 4 represents the structured and splined weather data of the corresponding energy data shown in Table 1. The description of column names of the weather data is provided in Table 5.

**Metadata.** Metadata is the information given about the data and plays a crucial role in connecting building energy data with other information such as weather, other characteristics of the buildings and corresponding analytics results.

**Table 4. Weather time series data structure.**

| tmst | temp | wspa | ghir | dhir | relh | gtir |
|------|------|------|------|------|------|------|
| 2016-10-15 00:00:00 | 5.27 | 2.67 | 0 | 0 | 82.59 | 0 |
| 2016-10-15 00:15:00 | 5.29 | 2.53 | 0 | 0 | 82.92 | 0 |
| 2016-10-15 00:30:00 | 5.31 | 3.03 | 0 | 0 | 82.68 | 0 |
| 2016-10-15 00:45:00 | 5.41 | 3.10 | 0 | 0 | 82.48 | 0 |

**Table 5. Column names and description of weather energy dataset.**

| column | column name | format | description |
|--------|-------------|--------|-------------|
| tmst | timestamp | Posixct | local time |
| temp | outside temperature | numeric | in ˚C |
| wspa | wind speed | numeric | in m/s |
| ghir | global horizontal irradiance | numeric | in $W/m^2$ |
| dhir | diffuse horizontal irradiance | numeric | in $W/m^2$ |
| relh | relative humidity | numeric | in % |
| gtir | global tilted irradiance | numeric | in $W/m^2$ |

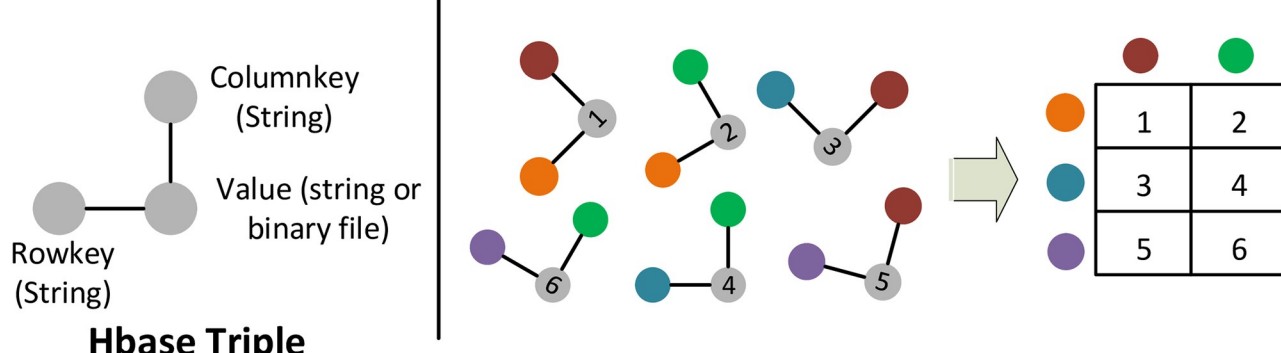

**Fig 4. HBase triples and their registration into a data table.** A rowkey and columnkey is assigned to each value. In the HBase data table triples share the same rowkey for a row and the columnkeys are the same for a column in data table.

- Metadata preparation: Building use type, number of floors, location, along with characteristics calculated from the building data and information derived from the initial dataset such as start and end time, and climate zone are stored in a dataframe and assigned to the buildings' de-identified name, and is then ingested into HBase.

- Metadata Security: The metadata can contain proprietary information about the building, which must be handled appropriately. For this we use a separate Research Electronic Data Capture (REDCap) database. REDCap is a patient tracking medical study database with HIPPA data privacy capabilities [51].

**Ingestion.** Triples for ingestion: HBase is a NoSQL database that operates under the Hadoop/HDFS distributed computing framework. It does not use a fixed table schema, as is typical for relational database management systems. For ingestion to HBase, a columnkey and rowkey are assigned to each value (Fig 4) [10, 52, 53]. An advantage of a non-relational data warehouse is that new variables, values and information can be added to the database tables without the need to refactor the table schemas. This ability to incorporate new table columns (new variables) and to have no performance impact of sparse columns is extremely important because as we analyze buildings and develop new building markers and analysis functions, writing back these results to HBase enables the overall dataset to be continually enhanced. In building-energy time-series data the *alphanumeric–yearmonth* of the dataset is considered as the rowkey and the column name as a column qualifier. And in each cell of a triple, we have a one month period of comma-separated datapoint values.

### Resources management

**High Performance Computing (HPC).** The Rider HPC cluster at Case Western Reserve University is a state of the art computing resource, used for large-scale, data intensive, computational problems. Three login nodes act as a gateway to the HPC environment, which consists of 4400 Intel Xeon compute cores. A separate SDLE Research Center's dedicated Hadoop, HBase, and Apache Spark [53, 54] cluster is integrated into the CWRU HPC environment. It consists of 180 compute cores, 2 TB of RAM, and 92 TB of disk space, configured as 12 data nodes, 2 name nodes, and 1 dedicated Apache Thrift and Rest server node [55]. We have developed R and Python packages that enable native interactions from either language to datasets stored in HBase tables by returning requested data as a dataframe in the R or Python environment.

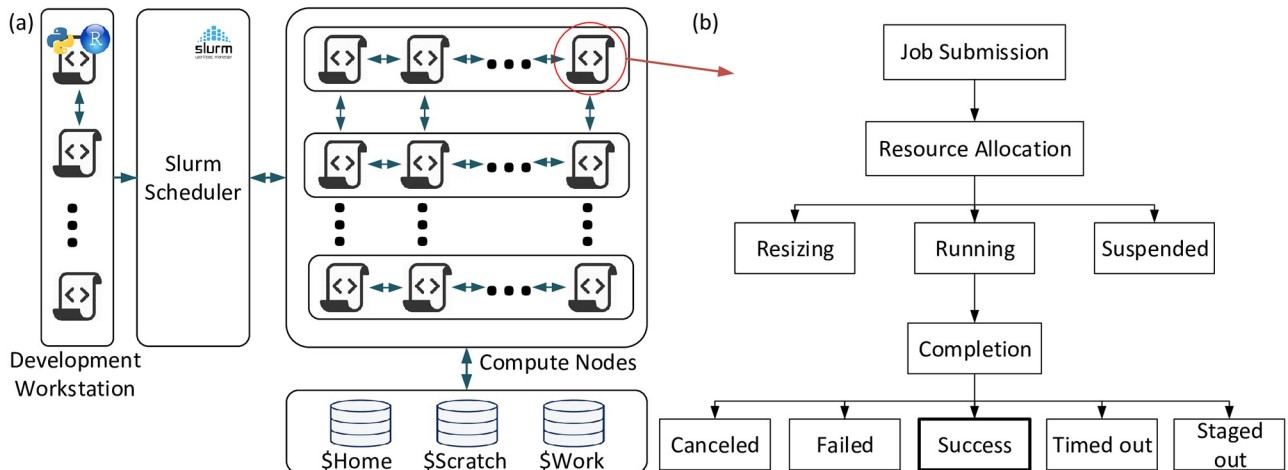

**Fig 5. Slurm jobs workflow and life cycle.** (a) jobs workflow and interactions with scheduler and storage and (b) job life cycle from submission to completion.

**Job schedulers.** The job scheduler contains four functions: life cycle management, resource management, scheduling and job execution [56]. These functions handle memory and accelerator allocation, licenses, prioritization and sorting of jobs, allocation of compute-nodes, job assignment to allocated resources, and reporting of logs.

**Slurm.** The Simple Linux Utility for Resource Management (Slurm), which was initially developed at the Lawrence Livermore National Laboratory, is a full-featured job scheduler with a multi-threaded core scheduler and substantially high scalability [57].

The Slurm workload manager is used to submit fleets of jobs in HPC [56] to speed up the process and improve fault tolerance. Fig 5a indicates how a Slurm scheduler takes jobs from a workstation that can be run through its cores and submits them to compute-nodes with robust specifications. The compute and login nodes have access to the storage environment of the home, scratch, and work directories. Completion of each job does not necessarily end up with "success" status. Fig 5b represents the lifecycle of a given Slurm job. As can be seen, there are several unsuccessful completion status for jobs, due to temperature or permanent issues. Therefore, monitoring and controlling the jobs throughout their life cycle is necessary for successfully completion of jobs and reporting their status.

The resources for Slurm jobs can be modified based on the dataset and computational intensity. The number of nodes and cores per node, memory, time, etc. can be controlled in the resource allocation step. This step is done using batch scripting. The flowchart of Slurm job controller that takes a script for a task and generates fleets of jobs for a population of data-sets is shown in Fig 6. After submitting each job, the resources are allocated. This step may be time-consuming and proper allocation of resources affects the speed of the process. Finally, the job runs until completion.

## Building-energy analytics pipeline

As shown in Fig 7, in building-energy analytics, HBase is queried for the required dataset, and upon retrieval it is transformed into an S3 R dataframe object [58] for analysis. Upon completion of building energy analysis, the results are stored as triples in the HBase results table. The results which are plain text are stored as text, while results consisting of binary items (such as model objects, plot file objects, or png files) which consist of multiple items and types, are

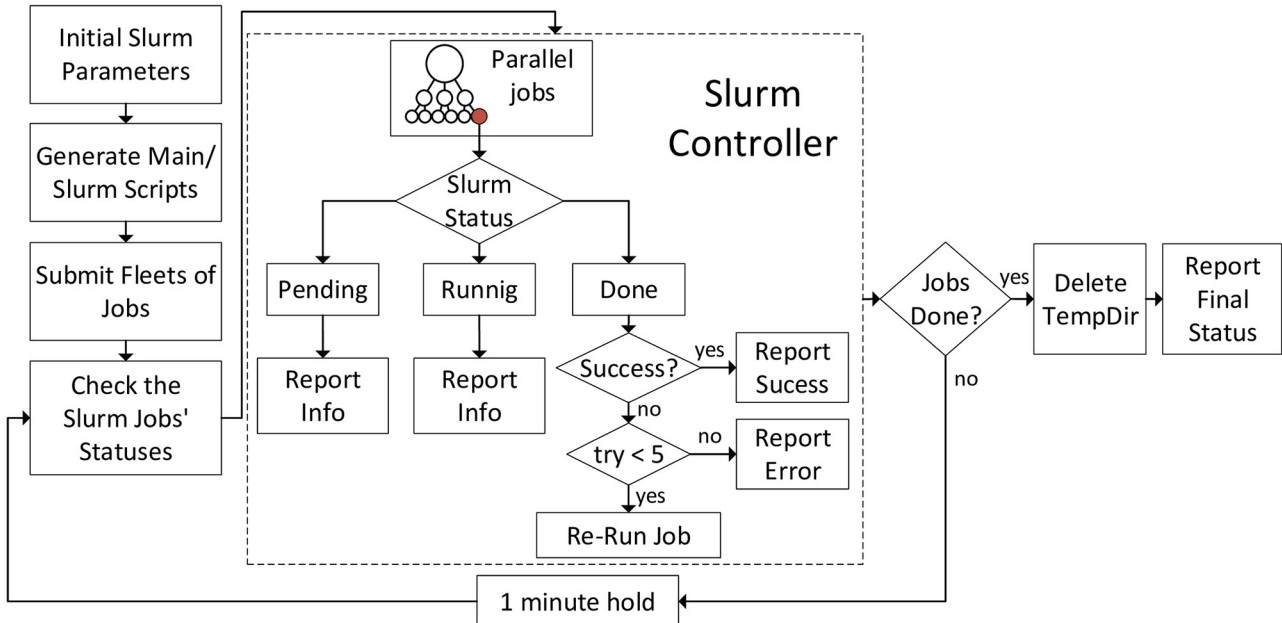

**Fig 6. Slurm job controller flowchart.** It represents the designed distribution and management of jobs and actions based on the execution result of each job.

combined in a single S3 R object and then stored as binary information in the HBase result table. If the results are dataframes, we transform them back to a packed cell text format for storage.

## Results

### Benchmarking

To benchmark the performance of our Building Energy Analytics Pipeline we compare analysis of 816 buildings in our population analyzed sequentially or using the pipeline. The analytics pipeline follows the steps presented in Fig 7, where in each job the data is queried from HBase, and the analysis is done and the results are stored back in the HBase results table. In the

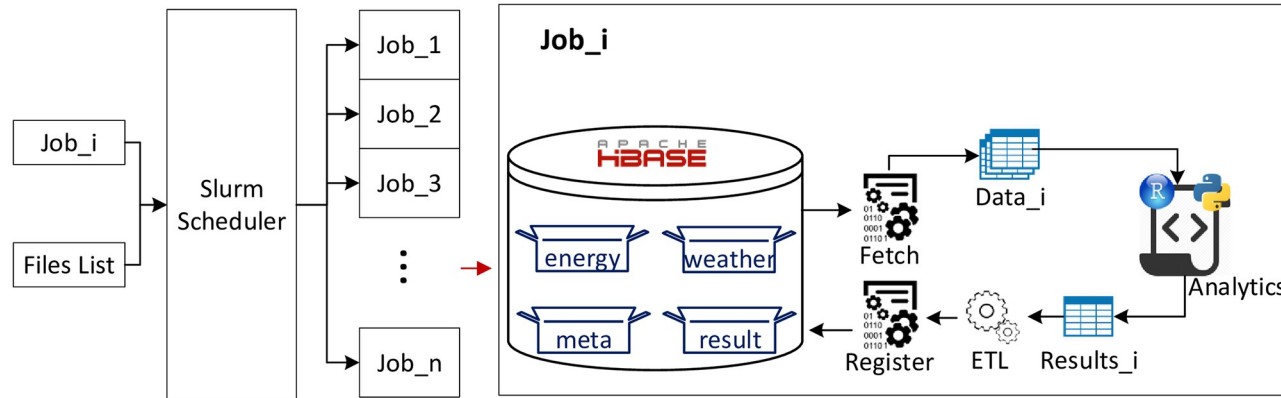

**Fig 7. Analytics workflow of jobs.** Jobs are distributed through Slurm scheduler and in each job data are fetched from HBase and converted to a dataframe. After being analyzed, the results are converted to HBase triples and registered to the results data table.

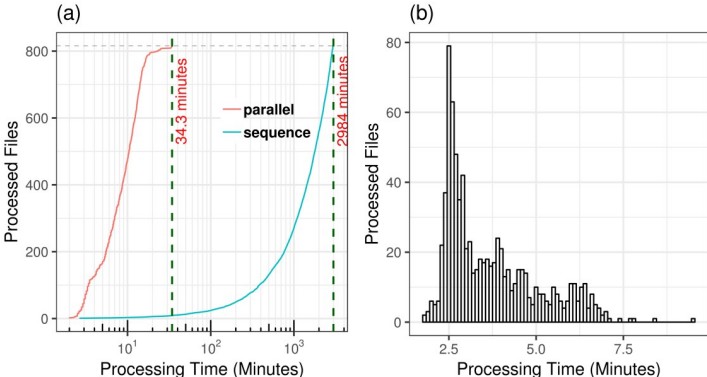

**Fig 8. Benchmarking of jobs.** (a) comparison of single-core parallel and sequential job processing times, (b) distribution of individual building analysis times for the 814 buildings in our population.

analytics, the completion time of each dataset is calculated. A comparison of the parallel and sequential job execution times and individual building analysis times are shown in Fig 8a. Post execution text processing of the job log files enables us to quantify the job timing and life cycle. In addition, the pipeline approach provides automated error checking and occurrence reporting to the user, an advantage over the sequential approach.

Each job can be assigned to multiple cores for less computational time (multi-core parallel jobs). However, since in parallel processing in a multi-core compute-node only one of the cores in each job can query the data from HBase, a significant portion of job execution time is consumed in the query and data i/o tasks. So, we do not expect significant savings in the amount of time with multi-core parallel job execution. For the validation of the best combination of cores in parallel Slurm job execution, we submitted the same set of buildings as analysis jobs to HPC with 1, 2, 4, and 8 cores for each job and compute-node. As our resources for each Slurm fleet is restricted to 120 cores per user, increasing the number of cores will lead to a reduction in the number of parallel nodes and thus reduction in number of jobs that can be executed simultaneously. Therefore, if we allocate one core per job, 120 nodes will work simultaneously, and with the allocation of two and four cores, we get 60 and 30 nodes running at the same time, respectively. Fig 9 demonstrates the performance of the pipeline with single-core and multi-core parallel job submission systems.

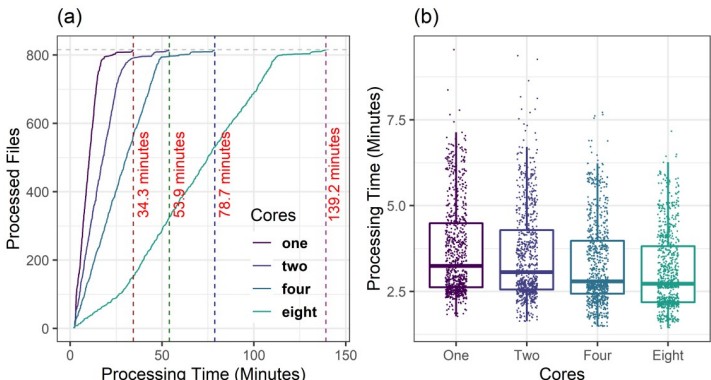

**Fig 9.** Processing time comparison of jobs with number of cores, (a) single-core and multi-core parallel jobs processing times, (b) performance of individual jobs with allocation of cores within each job.

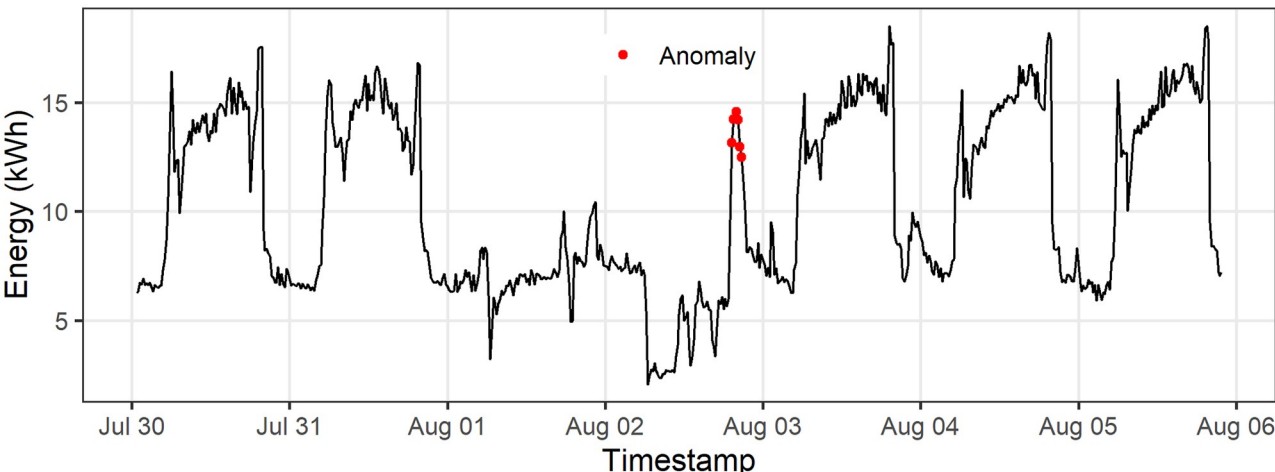

**Fig 10. Anomalies in energy consumption data.** Points represent the anomalies and the line represents the energy consumption in kWh.

### Data qualification tool

The data qualification tool identifies data cleaning issues such as missing datapoints, gaps, and anomalies. Then, it assigns a grade to the building energy dataset, and for the datasets with quality grade of lower than *AAAP*, submits the dataset for additional cleaning processes. Fig 10 represents anomalies in the time series data.

After data qualification and cleaning for the full 812 building population, the results shown in Fig 11 show that 40 buildings were upgraded to *AAAP*, leading to 752 high quality building energy datasets in the study population. In addition, 16 buildings failed with a final grade of *AAAF*, because the final time series was shortened to less than one year in length, making those buildings ineligible for analysis. Fig 11b shows the progressive improvement of the graded data quality by the sequential data cleaning processes of the data qualification tool.

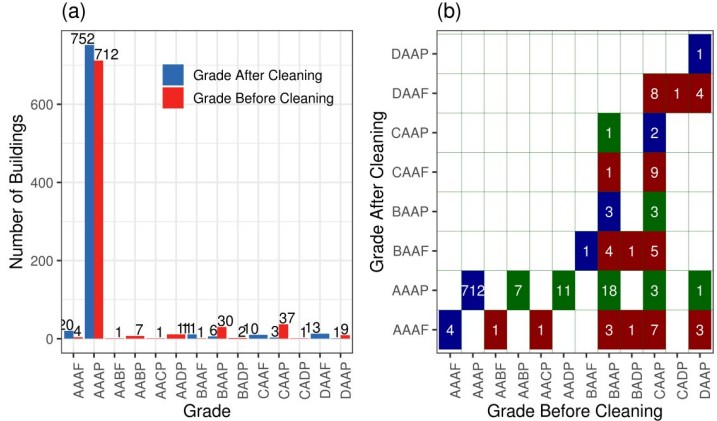

**Fig 11. Data quality population of data.** (a) Breakdown of data quality before and after cleaning (b) Status of data quality after cleaning. Blue represents the data without change in quality, red represents the data that failed after cleaning criteria, and green represents data that passed after cleaning.

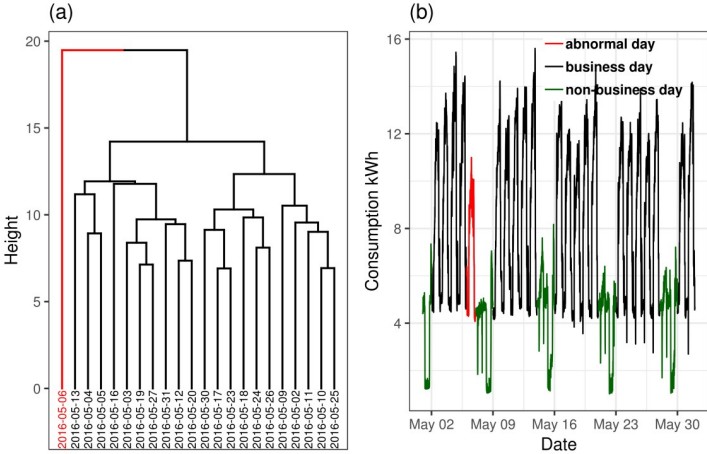

**Fig 12. Clustering on a month of data.** (a) Hierarchical clustering with red cluster showing the abnormal days, (b) Energy consumption plot of one month with abnormal days detection.

## Abnormal days

Some data quality issues do not arise from simple meter equipment data errors. We define these as abnormal days, such as days corresponding to extraordinary energy consumption, as could arise for a non-business day, where identifying them can enhance the interpretation of the results inferred from data. To identify abnormal days, we use hierarchical clustering [48] to classify the daily energy consumption during business days and identify and flag days with abnormal or uncharacteristic behavior such as high and low consumption. Fig 12a represents the clustering dendrogram of business days for one month of a building energy dataset, and in Fig 12b one can see the energy consumption curve of the clustered abnormal days compared to other typical business days.

## Population study: HVAC schedules across building types and climate zones

The automated Building Energy Analytics Pipeline enables populations of buildings to be studied to develop statistically significant results, as compared to a smaller set of buildings used in typical observational studies. As an example of this, we evaluated the HVAC turn-on and turn-off times of each building and compared the distribution of HVAC cooling on and off times across different building use types. We evaluated these for cooling degree days, which are the days during which the average daily temperature is above the thermostat setpoint of the cooling system that is required to operate. The specific HVAC turn-on and turn-off times represent the building's HVAC schedule, which is essential for identifying savings opportunities to reduce costs associated with air conditioning during relatively hot weather. The HVAC turn-on and off schedules across the population of buildings are broken out for the 10 different building use types, as shown in Fig 13.

The detailed distributions, or population densities for the HVAC turn-on and turn-off events of the ten building use types are presented in Table 6 and the median, interquartile range (IQR), and skewness are given. The median of the distribution is a measure of central tendency and is most relevant for normal distributions, as compared to bimodal or highly skewed distributions. IQR is a measure of the statistical dispersion and is equal to the difference of third and first quartiles, which is also a useful measure in normal distributions. Skewness is a measure of the asymmetry of the distribution, and can help identify both skewed normal distributions and bimodal or other distribution types. A distribution with positive

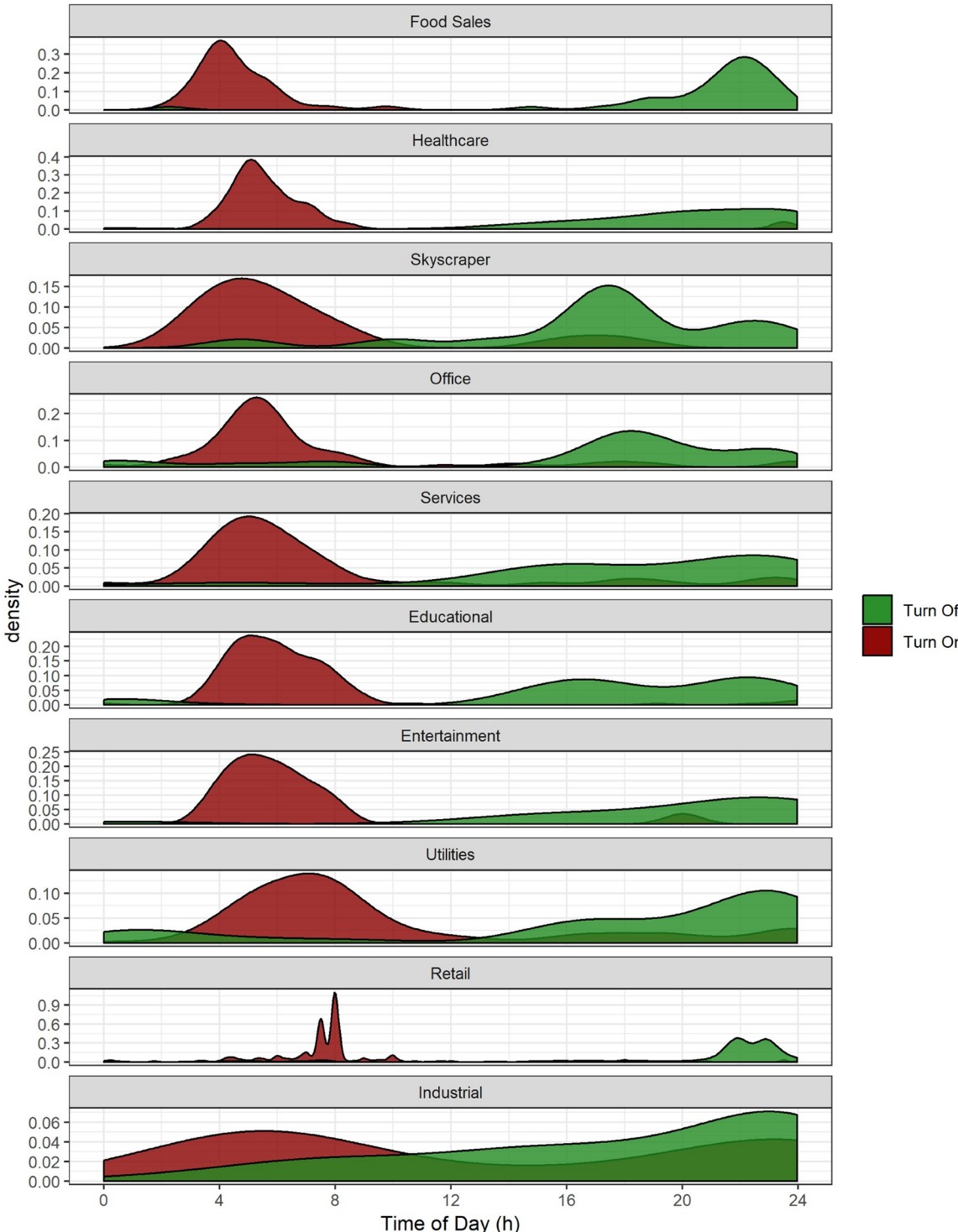

**Fig 13. Breakdown of HVAC scheduling time density on 10 different building use types.** Red and green colors represent turn on and turn off times, respectively.

**Table 6. Turn on and off times of HVAC systems of different building use types for cooling degree days.**

| | industry | turn on | | | turn off | | |
|---|---|---|---|---|---|---|---|
| | | median | IQR | skewness | median | IQR | skewness |
| 1 | Food Sales | 4.25 | 1.75 | 1.57 | 21.75 | 2.00 | -3.62 |
| 2 | Healthcare | 5.25 | 1.35 | 3.87 | 20.83 | 5.12 | -2.31 |
| 3 | Skyscraper | 5.50 | 3.19 | 1.54 | 17.75 | 3.21 | -0.94 |
| 4 | Office | 5.51 | 2.45 | 2.03 | 18.12 | 4.06 | -1.30 |
| 5 | Services | 5.75 | 3.00 | 1.74 | 20.67 | 7.72 | -1.26 |
| 6 | Educational | 5.97 | 2.29 | 3.62 | 18.00 | 6.05 | -1.64 |
| 7 | Entertainment | 5.97 | 1.75 | 2.77 | 21.75 | 7.25 | -1.88 |
| 8 | Utilities | 7.75 | 4.79 | 1.24 | 21.22 | 7.03 | -1.21 |
| 9 | Retail | 7.97 | 0.52 | 2.27 | 22.00 | 1.43 | -2.43 |
| 10 | Industrial | 9.77 | 18.38 | 0.15 | 19.93 | 9.87 | -0.73 |

skewness (right-skew) has a longer tail on the right side of the distribution, while negative skewness has the longer tail on the left side.

With the determination of the building schedule, the operating time of the cooling system is obtained, which is representative of expected occupancy. Fig 14 represents the breakdown of

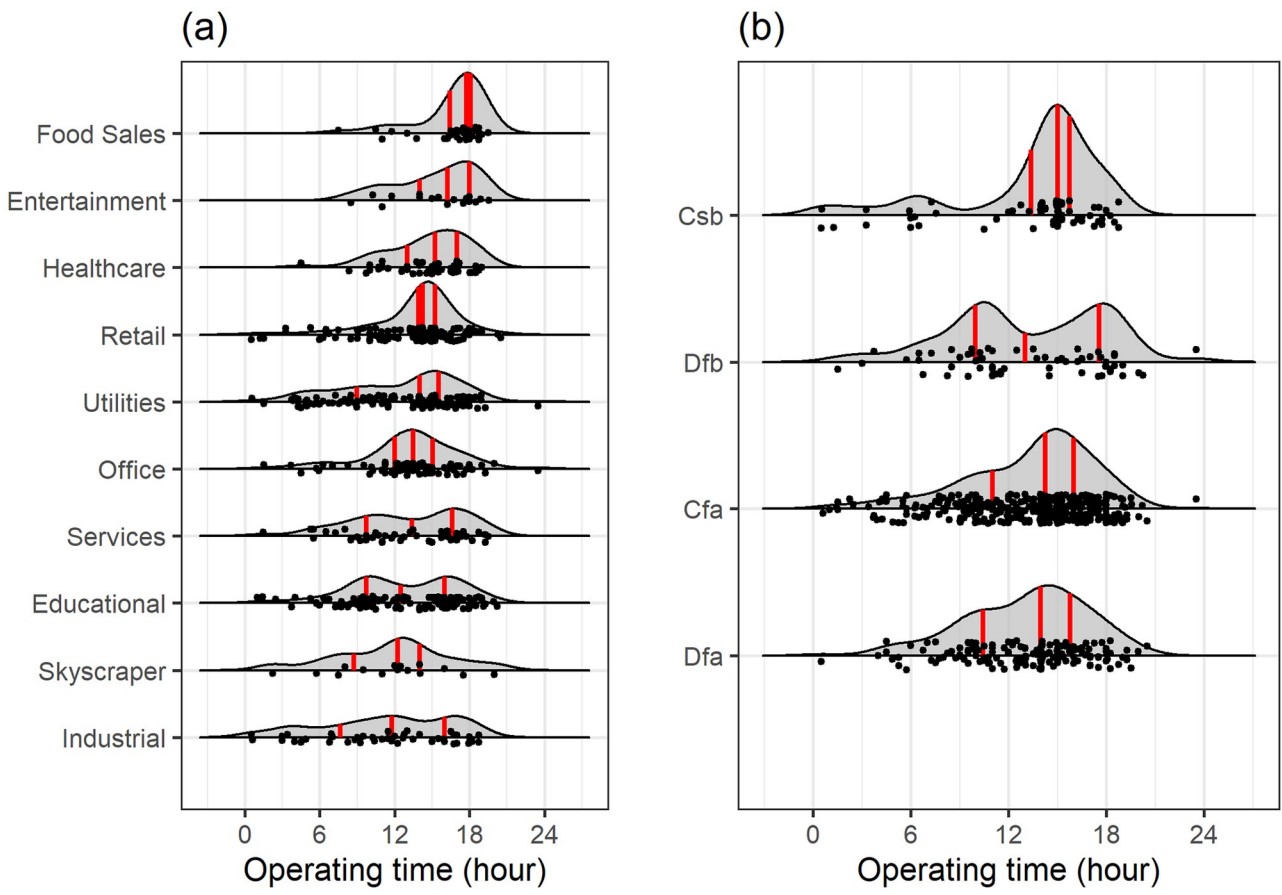

**Fig 14.** Breakdown of operating time for (a) 10 building use types and (b) four KG climate zones. Three red bars represent first, second and third quartiles. Dots represent the actual operating hour of each building and black lines represent the density of the distribution in each category.

operating time in different building use type and four KG climate zones for which we have statistically significant quantities of buildings in. The building use types are sorted from the ones with highest median operating time to the lowest.

## Discussion

### Data qualification

Anomalies, despite the possibility of representing physical meaning such as abrupt equipment turn on and off, alter our confidence in the analytical results. The Building Energy Analytics Pipeline's anomaly detection detects and flags anomalous data points. Fig 10 shows detection of an anomalous point due to an irregular spike. Further cleaning using the developed qualification tool can resolve most of the cleaning issues of the data. Applying the qualification tool on 816 buildings shown in Fig 11 results in improving the grade of 40 low quality data to *AAAP* with 18 buildings upgrading from *BAAP* to *AAAP*. Note that if the qualification tool could not improve the quality of data, the original format is still stored and can be analyzed.

Further improvement in the validity of analytics results is done by detecting anomalous days. Applying the hierarchical clustering algorithm to the data singles out the days with irregular patterns. For example, by applying the developed method on a month of data shown in Fig 12, it can be seen that the detected day, which is a Friday, has significantly lower consumption compared to other days.

### Parallel slurm jobs performance

The execution of fleets of parallel Slurm jobs in HPC lowers the computation time for large-scale data, dramatically. As shown in Fig 8a, the fleets of single-core parallel jobs process all the files in 34.3 minutes, which is 85 times faster than sequential execution. The slowing down of parallel jobs at the end of the execution is due to processing of the much larger, 1-minute interval, building energy datasets, which therefore require more data cleaning time, and failure of jobs due to temporary issues that are resolved with re-submission of jobs. As illustrated in Fig 8b, the majority of jobs are executed in around 2.5 minutes, while some jobs for larger datasets require more processing time. Overall, completion of all individual jobs takes less than 10 minutes.

Comparison of single-core and multi-core parallel Slurm jobs (Fig 9) shows that for large-scale data, single-core parallel Slurm jobs execution results in the lowest total processing time. With the implementation of two-core parallel Slurm jobs, the jobs are completed in 53.9 minutes, which is 1.57 times slower than single-core implementation. However, by increasing the number of cores per job, the completion time of individual jobs is faster (Fig 9b). Note that the allocation time of resources with increasing the number of cores per job increases due to restrictions in the availability of multiple cores in a compute-node.

For example, when a single core is assigned to a job, it can get the core from any compute-node. Even multiple jobs could get cores of the same compute-node. However, if eight cores are assigned to a job, a compute-node with 8 cores free for allocation is required which might not be available promptly, resulting in a waiting time.

### Population study: HVAC schedules across building type and climate zone

Utilization of the Building Energy Analytics Pipeline on a population of buildings enabled conducting a large-scale comparative study of HVAC schedules by building use type and KG climate zone. As illustrated in Fig 13, turn on schedules are more normally distributed with

less variability compared to turn off times. Also, the turn-on schedules are right-skewed with skewness ranging from 0.15 to 3.87, unlike turn-off schedules which are left-skewed.

The breakdown of the savings by building use types show that food sales buildings have the earliest turn-on schedules with a median of 4.25 (4:15 AM) and turn-off time with median of 21.75 (9:45 PM). Furthermore, the difference in skewness of the turn-on schedules illustrates that food sales have a tighter turn-on schedule. In healthcare buildings turn-on and off schedule patterns are completely different. The turn-on variability is one of the lowest in all buildings types with an IQR of 1.35 compared to 5.12 for turn off times, illustrating a much tighter control of the turn-on schedules. In industrial buildings the distributions of schedules are almost uniform, implying lack of schedule. Also, office, entertainment, and utility building use types have most spread in turn off times with IQRs of 7.72, 7.25, and 7.03 respectively. Despite less variability in turn-on times, utilities, skyscrapers, and services have the highest variability in turn on times with IQRs of 4.79, 3.19, and 3 respectively. Retail buildings have the lowest variability in both turn-on and turn-off times with IQRs of 0.52 and 1.43, implying a tight schedule.

Longer operating times lead to greater energy consumption offering more HVAC related savings opportunities. The results of the breakdown of operational hours by building use type (Fig 14) show that food sales buildings are a decent target for this purpose, with an operating time of 17.75 hours and a relatively small variability of 1.62. Also, the HVAC operating time of entertainment, healthcare, and retail buildings have the highest operating time of HVAC with that of retail buildings being relatively high. The bimodal distribution in educational buildings corresponds to two types of scheduling patterns in them. The scheduling in rest of the building use types, i.e. Utilities, Services, Skyscrapers, and industrial buildings are more uniformly distributed compared to other building use types. The evaluation of operating time in different climate zones represents a very similar distribution of Cfa and Dfa climate zones. This is because these climate-zones are in fully humid and hot summer areas, and the analytics results are for cooling degree days. It is represented in Fig 14 that the operating time and turn on/off schedules correlate more strongly with building use type compared to climate zone, since, the scheduling is more impacted by building management system.

## Conclusion

In this study, we introduced a fully automated Building Energy Analytics Pipeline for processing large volumes of building energy time series and developed a robust data cleaning process that not only improves the quality of dataset, but also, detects the daily consumption pattern and abnormalities. Our data qualification tool grades the quality of data in terms of anomalies, missing data points, and gaps, and if possible, improves the low-quality datasets to the highest standard with proper imputation and subsetting methods. In the processing of 816 buildings datasets with 712 of them already of high quality, we were able to upgrade the quality of 40 low-quality data to the highest grade of *AAAP*.

The processed and analyzed datasets along with meta and weather data are transformed into HBase triples and ingested into HBase for analysis. This pipeline and associated compute infrastructure is capable of fast dataset processing at scale with robust error handling. For optimal allocation of computational resources, we also designed a smart Slurm scheduler on top of the HPC Slurm infrastructure, that controls all the jobs and manages their lifecycle. Our pipeline can process 816 buildings in less than 35 minutes which is 85 times faster than a sequence processing time.

In addition to data anamolies, the dataset quality issues as a result of abnormal time series patterns arising from irregular equipment operation are also identified. For example, with

hierarchical clustering, days with abnormal time series pattern arising from significantly high, low, or irregular consumption are detected. The abnormal days are flagged in the time series data for exclusion or inclusion in further analysis.

By utilizing the BEA pipeline, we are able to analyze the HVAC cooling schedule of a population of 816 buildings', broken out into 10 building use types and 4 KG climate zones with a statistically significant number of buildings. We compared the HVAC performance with turn on and turn off schedules, discussed their distribution and identified high potential building use types for savings. The results show that food sales buildings have the highest air conditioning operational hours (a median of 17 hours) and thus are decent targets for savings. Also, the retail buildings have the least variability in their schedule with turn on and turn off IQRs of 0.53 and 1.43, respectively. The breakdown of scheduled hours by climate zones showed that the Cfa and Dfa climate zones have a similar distribution of operating time. The results showed that the operating period correlates with building use types stronger than climate zones. Our developed pipeline addresses the automation, scalability and efficiency challenges of large-scale time-series processing, hence, can be utilized in live buildings data analysis applications.

## Supporting information

**S1 Text.**
(TXT)

## Acknowledgments

This research was performed in the SDLE Research Center, established with Ohio Third Frontier funding under award Tech 11-060, Tech 12-004, and the Great Lakes Energy Institute, both at Case Western Reserve University. This work made use of the Rider High Performance Computing Resource in the Core Facility for Advanced Research Computing at Case Western Reserve University. The authors acknowledge useful discussions with Tian Wang.

## Author Contributions

**Conceptualization:** Arash Khalilnejad, Ahmad M. Karimi, Rojiar Haddadian.

**Data curation:** Arash Khalilnejad.

**Formal analysis:** Arash Khalilnejad.

**Investigation:** Arash Khalilnejad.

**Methodology:** Arash Khalilnejad, Shreyas Kamath, Rojiar Haddadian, Roger H. French.

**Project administration:** Arash Khalilnejad, Roger H. French, Alexis R. Abramson.

**Resources:** Arash Khalilnejad, Roger H. French.

**Software:** Arash Khalilnejad.

**Supervision:** Roger H. French, Alexis R. Abramson.

**Validation:** Arash Khalilnejad.

**Visualization:** Arash Khalilnejad.

**Writing – original draft:** Arash Khalilnejad.

**Writing – review & editing:** Arash Khalilnejad, Ahmad M. Karimi, Shreyas Kamath, Rojiar Haddadian, Roger H. French.

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
