## [Decision Letter · Decision Letter 0]

6 May 2020

PONE-D-20-05132

Automated pipeline framework for processing of large-scale building energy time series data

PLOS ONE

Dear Professor French,

Thank you for submitting your manuscript to PLOS ONE. After careful consideration, we feel that it has merit but does not fully meet PLOS ONE’s publication criteria as it currently stands. Therefore, we invite you to submit a revised version of the manuscript that addresses the points raised during the review process.

We would appreciate receiving your revised manuscript by Jun 20 2020 11:59PM. To enhance the reproducibility of your results, we recommend that if applicable you deposit your laboratory protocols in protocols.io, where a protocol can be assigned its own identifier (DOI) such that it can be cited independently in the future. For instructions see: http://journals.plos.org/plosone/s/submission-guidelines#loc-laboratory-protocols

We look forward to receiving your revised manuscript.

Kind regards,

Jian Shen

Academic Editor

PLOS ONE

Journal Requirements:

2. Please amend your list of authors on the manuscript to ensure that each author is linked to an affiliation. Authors’ affiliations should reflect the institution where the work was done (if authors moved subsequently, you can also list the new affiliation stating “current affiliation:….” as necessary).

Reviewers' comments:

Reviewer's Responses to Questions

**Comments to the Author**

1. Is the manuscript technically sound, and do the data support the conclusions?

Reviewer #1: Yes

Reviewer #2: Yes

2. Has the statistical analysis been performed appropriately and rigorously? 

Reviewer #1: Yes

Reviewer #2: Yes

3. Have the authors made all data underlying the findings in their manuscript fully available?

Reviewer #1: Yes

Reviewer #2: Yes

4. Is the manuscript presented in an intelligible fashion and written in standard English?

Reviewer #1: Yes

Reviewer #2: Yes

5. Review Comments to the Author

Reviewer #1: The manuscript demonstrates a computing method that claims to be faster than others in order to sort and clean building energy data. The manuscript is generally well written and the topic of study is within the scope of the journal. It is suggested accepting this paper after minor revisions concerning the following comments and questions.

1. In the abstract, it is mentioned that this pipeline can analyze the data. In the body of the paper, there is no discussed methodology on how to analyze energy data. For example, HVAC has factors, such as the fan speed, coils, return air temperature, space temperature, discharge air temperature and others that need to be considered and weighted to analyze HVACs. It is suggested to just mention data cleaning and sorting at this stage.

2. In the introduction, last paragraph, “data automatically goes through multiple processing”, how data automatically goes through the process?

3. It is suggested to explain if this pipeline can sort data live-online with connecting to some global controls, or like some other commissioning software, data need to be uploaded. Having high-speed computing capability along with live-online data analyzing tools are considerable value propositions.

4. The author should consider explaining his definition of automated analytics.

5. Most figures have a long description. It is suggested to shorten the titles and add explanations to the text.

Reviewer #2: This paper proposes an automated pipeline building energy analytics -- BEA --a pipeline for virtual energy audit applied to time series. Authors use data sets in a non relational data warehouse in high-performance computing to manage job scheduling for a parallel processing. An interesting contribution is the protocol to assemble datasets and prepare for analysis. It also includes tools for data qualification that enhances data quality, and a machine learning algorithm: hierarchical clustering. The reported scalability is impressive.

Other suggestion:

* line 50: change has lead to has led

6. PLOS authors have the option to publish the peer review history of their article (what does this mean?). If published, this will include your full peer review and any attached files.

Reviewer #1: No

Reviewer #2: No

---

## [Author Response · Author response to Decision Letter 0]

23 Jul 2020

Reviewer #1: The manuscript demonstrates a computing method that claims to be faster than others in order to sort and clean building energy data. The manuscript is generally well written and the topic of study is within the scope of the journal. It is suggested accepting this paper after minor revisions concerning the following comments and questions.

1. In the abstract, it is mentioned that this pipeline can analyze the data. In the body of the paper, there is no discussed methodology on how to analyze energy data. For example, HVAC has factors, such as the fan speed, coils, return air temperature, space temperature, discharge air temperature and others that need to be considered and weighted to analyze HVACs. It is suggested to just mention data cleaning and sorting at this stage.

We added a sentence to clarify that the pipeline can enable any data-driven analysis, and we have shown its performance with analyzing the HVAC scheduling. 

“Our developed pipeline enables any applied analysis to be distributed to all the datasets and results stored in the database, automatically.”

2. In the introduction, last paragraph, “data automatically goes through multiple processing”, how data automatically goes through the process?

The use of “automatically goes through the process” is revised in the paper and now reads:

“In this paper, we demonstrate the development of an energy analytics pipeline wherein data, after being queried, automatically passes through multiple preprocessing, cleaning, assembly and ingestion steps in high performance and parallel computing environment with fast-track, smart and interactive capabilities.”

3. It is suggested to explain if this pipeline can sort data live-online with connecting to some global controls, or like some other commissioning software, data need to be uploaded. Having high-speed computing capability along with live-online data analyzing tools are considerable value propositions.

The reviewer points out there is increased value sith capability of live-online data analyzing, which is very true, and our pipeline enables such a capability. We elaborated it as:

“Our developed pipeline addresses the automation, scalability and efficiency challenges of large-scale time-series processing, hence, can be utilized in live buildings data analysis applications.”

4. The author should consider explaining his definition of automated analytics.

An explanation regarding automated analytics has been added,

“The automated process which implies the automatic flow of building data to each step of the pipeline, is designed to do all the required steps for data processing and analytics automatically with an efficient framework and unified structure, and generate comparable results with data from different sources and formats.”

5. Most figures have a long description. It is suggested to shorten the titles and add explanations to the text.

According to the recommendation of the reviewer, we updated the explanations of some of the figures.

Reviewer #2: This paper proposes an automated pipeline building energy analytics -- BEA --a pipeline for virtual energy audit applied to time series. Authors use data sets in a non relational data warehouse in high-performance computing to manage job scheduling for a parallel processing. An interesting contribution is the protocol to assemble datasets and prepare for analysis. It also includes tools for data qualification that enhances data quality, and a machine learning algorithm: hierarchical clustering. The reported scalability is impressive.

---

## [Decision Letter · Decision Letter 1]

28 Sep 2020

Automated pipeline framework for processing of large-scale building energy time series data

PONE-D-20-05132R1

Dear Dr. French,

We’re pleased to inform you that your manuscript has been judged scientifically suitable for publication and will be formally accepted for publication once it meets all outstanding technical requirements.

Kind regards,

Rashid Mehmood, PhD

Academic Editor

PLOS ONE

Additional Editor Comments (optional):

Reviewers' comments:

Reviewer's Responses to Questions

**Comments to the Author**

1. If the authors have adequately addressed your comments raised in a previous round of review and you feel that this manuscript is now acceptable for publication, you may indicate that here to bypass the “Comments to the Author” section, enter your conflict of interest statement in the “Confidential to Editor” section, and submit your "Accept" recommendation.

Reviewer #1: All comments have been addressed

2. Is the manuscript technically sound, and do the data support the conclusions?

Reviewer #1: Yes

3. Has the statistical analysis been performed appropriately and rigorously? 

Reviewer #1: Yes

4. Have the authors made all data underlying the findings in their manuscript fully available?

Reviewer #1: Yes

5. Is the manuscript presented in an intelligible fashion and written in standard English?

Reviewer #1: Yes

6. Review Comments to the Author

Reviewer #1: The authors have addressed reviewer's comments in the revised version. The manuscript can be accepted for publication.

7. PLOS authors have the option to publish the peer review history of their article (what does this mean?). If published, this will include your full peer review and any attached files.

Reviewer #1: No

---

## [Editor Report · Acceptance letter]

21 Oct 2020

PONE-D-20-05132R1 

Automated pipeline framework for processing of large-scale building energy time series data 

Dear Dr. French:

I'm pleased to inform you that your manuscript has been deemed suitable for publication in PLOS ONE. Congratulations! Your manuscript is now with our production department. 

Kind regards, 

on behalf of

Dr. Rashid Mehmood 

Academic Editor

PLOS ONE